# RARE EVENT MODELING WITH SELF-REGULARIZED NORMALIZING FLOWS: WHAT CAN WE LEARN FROM A SINGLE FAILURE?

**Charles Dawson**
Department of Aeronautics and Astronautics
Massachusetts Institute of Technology
cbd@mit.edu

**Van Tran**
Department of Applied Mathematics
Harvard University
vantran@college.harvard.edu

**Max Z. Li**
Department of Aerospace Engineering
University of Michigan
maxzli@umich.edu

**Chuchu Fan**
Department of Aeronautics and Astronautics
Massachusetts Institute of Technology
chuchu@mit.edu

## ABSTRACT

Increased deployment of autonomous systems in fields like transportation and robotics has led to a corresponding increase in safety-critical failures. These failures are difficult to model and debug due to the relative lack of data: while normal operations provide tens of thousands of examples, we may have only seconds of data leading up to the failure. This scarcity makes it challenging to train generative models of rare failure events, as existing methods risk either overfitting to noise in the limited failure dataset or underfitting due to an overly strong prior. We address this challenge with CALNF, or calibrated normalizing flows, a self-regularized framework for posterior learning from limited data. CALNF achieves state-of-the-art performance on data-limited failure modeling problems and enables a first-of-a-kind case study of the 2022 Southwest Airlines scheduling crisis.

## 1 INTRODUCTION

When complex systems fail, the first step towards recovery is understanding the factors that lead to failure. Modeling failures and other rare events is challenging because the limited amount of available data. While much work has been done on preemptive failure prediction in simulation (Corso et al., 2022; O' Kelly et al., 2018; Sinha et al., 2020; Dawson & Fan, 2023; Delecki et al., 2023; Zhong et al., 2023), and online failure detection (Keipour et al., 2021; Hendrycks et al., 2018; Gudovskiy et al., 2022; Kang et al., 2022; Najari et al., 2022; Garg et al., 2023), relatively little work has been done on post-event failure modeling from observational data.

In failure modeling problems, we seek to infer the hidden factors contributing to an observed failure by learning a posterior distribution over those factors. When observational data are plentiful, deep generative models can be powerful tools for solving these problems, but these methods struggle when only a few examples are available. To solve rare event modeling problems that arise in domains like robotics and networked cyberphysical systems, we need new ways of training generative models in data-constrained settings.

Formally, we frame rare event modeling as a *data-constrained Bayesian inverse problem* where we aim to infer the distribution of latent variables $z$ from noisy observations $x$ of a stochastic process $x \sim p_\theta(x|z; y)$, where $\theta$ are unknown process parameters and $y$ are known context variables, all real-valued (Stuart, 2010). Given nominal observations $\mathcal{D}_0 = \{x_0^{(i)}, y_0^{(i)}\}_{i=1}^{N_0}$ and a much smaller set of target observations $\mathcal{D}_t = \{x_t^{(i)}, y_t^{(i)}\}_{i=1}^{N_t}$, where $N_t \ll N_0$, we aim to learn an approximation of the posterior distribution

$$q_\phi(z) \approx p_\theta \left( z | \{x_t^{(i)}, y_t^{(i)}\}_{i=1}^{N_t} \right). \tag{1}$$

Note that this problem is distinct from the rare event simulation problem considered in O' Kelly et al. (2018); Sinha et al. (2020); Gao et al. (2023), and Dawson & Fan (2023). While we focus on learning the distribution of failures from small, fixed number of real-world data points, these methods assume the ability to sample and label an arbitrary number of new failure examples in simulation.

In data-constrained settings, a common approach to solving inverse and few-shot generative modeling problems is to use the nominal observations to train a deep model of the prior distribution, then use this prior to regularize $q_\phi$ (e.g. by penalizing the divergence between $q_\phi$ and the learned prior, Asim et al. (2020); Liu et al. (2023); Abdollahzadeh et al. (2023); Zhang et al. (2019); Ojha et al. (2021); Higgins et al. (2016)). Unfortunately, it can be difficult to specify the appropriate amount of regularization *a priori*, particularly when the distribution of $z$ differs between the nominal and target datasets (as if often the case in failure modeling problems).

In this paper, we address this challenge by developing CALNF, or calibrated normalizing flows. To make full use of available data, CALNF amortizes inference over both the nominal and target datasets, learning a shared representation for both posteriors. To prevent overfitting, CALNF first learns a low-dimensional embedding for a family of candidate posteriors then searches over this low-dimensional space to find an optimal representation of the target posterior. Our method achieves state-of-the-art performance on a range of data-constrained inference benchmarks, including several with real-world data from autonomous systems. We apply our method to a post-mortem analysis of the 2022 Southwest Airlines scheduling crisis, which stranded more than 2 million passengers and led to more than \$750 million in financial losses (Rose, 2023); our first-of-a-kind analysis suggests a mechanism by which failure propagated through the Southwest network.

## 2 BACKGROUND

Because the posterior in Eq. (1) is typically intractable to evaluate exactly, inverse problems are typically solved approximately, often using variational inference (Stuart, 2010). These methods approximate the true posterior by maximizing the evidence lower bound (ELBO, Kingma & Welling (2013)), often using deep non-parametric representations like normalizing flows for $q_\phi$ (Tabak & Vanden-Eijnden, 2010; Rezende & Mohamed, 2015).

$$\mathcal{L}(\phi, \theta, \mathcal{D}) = \mathop{\mathbb{E}}_{(x,y)\in\mathcal{D}} \mathop{\mathbb{E}}_{z\sim q_\phi(z)} \left[\log p_\theta(x, z; y) - \log q_\phi(z)\right]. \quad (2)$$

A particular challenge in data-constrained settings is that deep models tend to overfit the particular target samples used for training, as shown in Fig. 1c. As the following result illustrates, the representational power of these models makes them highly sensitive when trained on small datasets.

**Lemma 1.** *Let $\mathcal{D} = \{z^{(i)}\}_{i=1}^N$ be sparse dataset with distance $O\left((LN)^{-1/(d+1)}\right)$ between points (the precise limit is given in the appendix), and let $q_\phi(z)$ be a model capable of representing any $L$-Lipschitz probability density. If $\phi(\mathcal{D})$ are the parameters of the maximum likelihood estimator[1] given $\mathcal{D}$, then the optimal solutions trained on datasets differing by one point $\mathcal{D}_1 = \mathcal{D} \cup \{z^{(1)}\}$ and $\mathcal{D}_2 = \mathcal{D} \cup \{z^{(2)}\}$ will differ by Wasserstein distance $W_2\left(q_{\phi(\mathcal{D}_1)}, q_{\phi(\mathcal{D}_2)}\right) = ||z^{(1)} - z^{(2)}||/N$.*

---

[1]We provide these results for maximum likelihood estimation, but similar results can be shown for maximum *a posteriori* or maximum ELBO contexts.

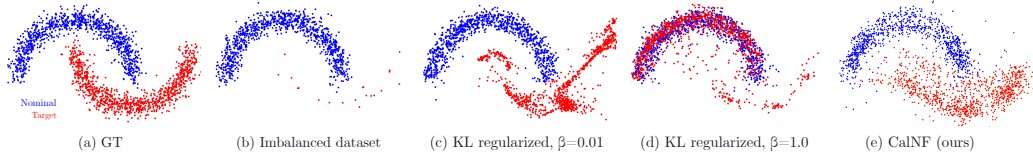

(a) GT     (b) Imbalanced dataset     (c) KL regularized, β=0.01     (d) KL regularized, β=1.0     (e) CalNF (ours)

Figure 1: **Inference in data-constrained environments.** (a) The ground truth distribution. (b) An imbalanced dataset. (c) When the regularization strength $\beta$ is too small, deep models overfit to noise in the target dataset. (d) When $\beta$ is too large, the learned distribution underfits and struggles to distinguish between nominal and target distributions. (e) Our method learns a more accurate reconstruction of the target distribution using hyperparameter-insensitive self-regularization.

The proof relies on the fact that the optimal $L$-Lipschitz maximum likelihood estimator is $q_{\phi(\mathcal{D})}(z) = \sum_{z^{(i)} \in \mathcal{D}} \hat{\delta}(z - z^{(i)})$, where $\hat{\delta}(z) = \max(0, a - L||z||)$ is an $L$-Lipschitz approximation of $\delta(z)/N$ and $a$ is a constant given in the appendix. Lemma 1 implies that small changes in the training data can lead to large changes in the learned distribution; as $N$ becomes small, this sensitivity increases.

Two common strategies for reducing this sensitivity in non-sparse settings are *bootstrapping* (Efron, 1992), which trains an ensemble of models on random subsamples of the target data, and *prior regularization* (Asim et al., 2020), which uses a model trained on nominal data to regularize the target posterior. Unfortunately, both of these methods are difficult to apply to data-constrained problems.

Bootstrapping, or ensemble, methods are theoretically well-motivated (Efron, 1992; Breiman, 1996) but do not fully avoid data sensitivity issues when applied to deep models (Nixon et al., 2020). In fact, it can be shown that bootstrapping has no effect on data sensitivity in highly data-constrained environments.

**Lemma 2.** *Consider the setting from Lemma 1. Let $\mathcal{D}_i$ be $K$ subsampled datasets created by sampling $N$ points with replacement from the original $\mathcal{D}$. Let $\phi(\mathcal{D}_i)$ be parameters of the maximum likelihood estimator for each $\mathcal{D}_i$ and define the ensemble model $q_{ensemble}(z) = \sum_{i=1}^{K} q_{\phi_i}(z)/K$. As $K \to \infty$, the ensemble model recovers the solution of the non-bootstrapped problem; i.e. $q_{ensemble}(z) \to q_{\phi(\mathcal{D})}(z)$.*

The second strategy, prior regularization, is common in the few-shot learning literature, particularly for image generation tasks (Asim et al., 2020; Higgins et al., 2016; Abdollahzadeh et al., 2023). These methods use a large dataset to learn the nominal posterior distribution, then use that nominal posterior to regularize the target posterior:

$$\phi_0 = \operatorname{argmax}_\phi \mathcal{L}(\phi, \theta, \mathcal{D}_0) \qquad \phi_t = \operatorname{argmax}_\phi \mathcal{L}(\phi, \theta, \mathcal{D}_t) - \beta D_{KL}(q_{\phi_0}, q_\phi) \qquad (3)$$

This penalty allows the large number of nominal samples to regularize the distribution learned from the target data, but there are two issues with this approach. First, it is difficult to choose an appropriate penalty strength $\beta$ *a priori*, as illustrated using the toy example in Fig. 1. If $\beta$ is too small, the model will overfit to the target data (Fig. 1c), but if $\beta$ is too large then the model will underfit the target in favor of learning the nominal distribution (Fig. 1d). Moreover, even if we were able to select an optimal $\beta$, many practical failure modeling problems involve a large shift between the nominal and target distributions, in which case regularizing between these distributions may not be appropriate.

In order to effectively train deep generative models of rare failure events, we will need to address two key questions. First, building upon prior regularization: can we adapt the amount of regularization based on the available data rather than specifying a regularization penalty *a priori*? Second, building upon bootstrapping: can we share information between model components to learn robustly in data-constrained contexts?

## 2.1 RELATED WORK

A number of recent works have explored various forms of prior regularization, mostly in the context of image generation tasks (see Abdollahzadeh et al. (2023) for a survey). Asim et al. (2020) use a deep prior model trained on large image datasets to regularize single-shot maximum likelihood estimation for image denoising. Liu et al. (2023) combine prior regularization with graduated optimization for image reconstruction, gradually increasing the regularization strength over multiple inference rounds. Ojha et al. (2021) pre-train a model on a large open image dataset, then fine-tune to smaller image datasets with a loss function that preserves relative difference and similarity across domains. Kong et al. (2022) focus on generative adversarial networks (GANs) and regularize the generator's latent space by sampling points in between sparse target data. Other works rely on data augmentation, either using hand-designed heuristics Wang et al. (2020), which must be re-derived for each new problem domain, or pre-trained generative models Tran et al. (2017); Zheng et al. (2023), which require a minimum amount of data from similar examples.

## 3 METHOD: CALIBRATED NORMALIZING FLOWS

To address the challenges from Section 2, we propose a novel framework for rare event modeling: calibrated normalizing flows, or CALNF. This method involves two steps, illustrated in Fig. 2. First, we learn a low-dimensional embedding for a family of probability distributions that includes both the

nominal posterior and several candidate target posteriors. We then search over this low-dimensional space to find the optimal representation of the target posterior. To ensure that the embedding space is well behaved, we apply regularization between candidate target posteriors. This self-regularization, combined with implicit regularization from our use of a single, shared representation for both the target and nominal posteriors, allows CALNF to efficiently learn the posterior without overfitting.

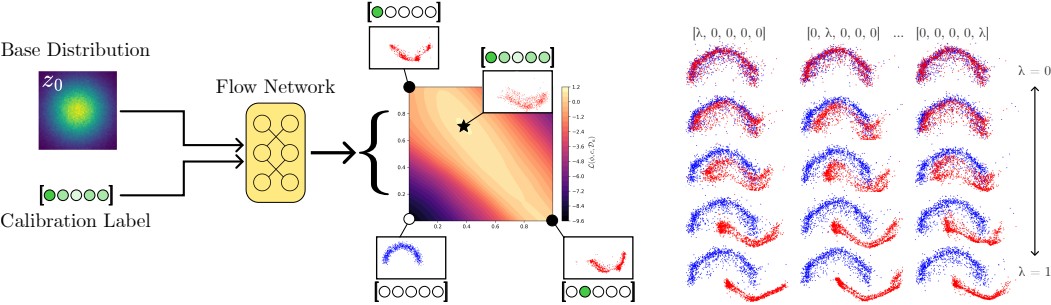

Figure 2: **(Left) CalNF architecture**: A normalizing flow is trained on random subsamples of the target data and the full nominal dataset, using one-hot labels to identify different subsamples (●) and the zero vector to identify the nominal data (○). The model is calibrated by freezing the model parameters and optimizing the label on the entire target dataset (★). **(Right) Target candidates**: The nominal posterior $q_\phi(z; \mathbf{0})$ (blue) and the family of candidate distributions for the target posterior $q_\phi(z; \lambda \mathbf{1}_i)$, shown for varying values of the calibration label.

The first step in our framework is to learn an embedding for a family of candidate probability distributions. To do this, we randomly sample $K$ subsets of the target data $\mathcal{D}_t^{(1)}, \ldots, \mathcal{D}_t^{(K)}$, then train a single conditional normalizing flow $q_\phi(z; c)$ to learn a mapping from a low-dimensional label $c$ to the posterior distribution conditioned on each of these subsets. We identify the posterior for each target subset using one-hot labels $\mathbf{1}_i$ and use the zero label $\mathbf{0}_K$ to identify the nominal posterior; i.e.:

$$q_\phi(z; \mathbf{0}_K) \approx p(z|\mathcal{D}_0), \qquad q_\phi(z; \mathbf{1}_i) \approx p(z|\mathcal{D}_t^{(i)}), \ i = 1, \ldots, K$$

Once posteriors have been learned for each of these subsets, we calibrate the model by holding the model weights $\phi$ constant and searching for an optimal label $c^*$ such that $q_\phi(z; c^*) \approx p(z|\mathcal{D}_t)$.

This training process is shown in more detail in Algorithm 1. This algorithm modifies the standard variational inference training process in two ways: by training on multiple random subsets of the target data, and by interleaving model updates and label calibration. To begin, we split the target training data into $K$ random subsets with one-hot labels and train the model to learn the posterior for each subset. Each subset $\mathcal{D}_t^i$ is created by independently drawing $N_t$ samples from $\mathcal{D}_t$ with replacement. We denote the empirical ELBO given dataset $\mathcal{D}$ as

$$\mathcal{L}(\phi, c, \mathcal{D}) = \frac{1}{|\mathcal{D}|} \sum_{(x,y) \in \mathcal{D}} \mathbb{E}_{z \sim q_\phi(z;c)} \left[ \log p(x, z; y) - \log q_\phi(z; c) \right]. \tag{4}$$

The model is trained with three objectives: to maximize the ELBO on each target subset (with one-hot labels), the nominal data (with a zero label), and the full target dataset (using the calibrated label $c^*$). In addition, to improve the conditioning of the embedding space, we add a regularization term on the divergence between the posterior distributions learned for each target subset, yielding the loss:

$$L(\phi, c) = -\frac{1}{K} \sum_{i=1}^{K} \mathcal{L}\left(\phi, \mathbf{1}_i, \mathcal{D}_t^{(i)}\right) - \mathcal{L}(\phi, \mathbf{0}_K, \mathcal{D}_0) - \mathcal{L}(\phi, c, \mathcal{D}_t) + \beta \sum_{i \neq j}^{K} D_{KL}\left(q_\phi(\cdot; \mathbf{1}_i), q_\phi(\cdot; \mathbf{1}_j)\right) \tag{5}$$

This additional regularization term reflects the fact that the target subsamples $\mathcal{D}_t^{(i)}$ are identically distributed, so the divergence between the candidate posteriors learned for each subsample should be small, and we show empirically that CALNF performs well for a range of $\beta$. The mixture label $c^*$ is initialized at $[1/K, \ldots, 1/K]$ and updated to maximize $c^* = \arg\max_c \mathcal{L}(\phi, c, \mathcal{D}_t)$. In practice, this is equivalent to jointly optimizing $(\phi, c^*) = \arg\min L(\phi, c^*)$, but we show these optimization steps separately in Alg. 1 to emphasize the different objectives.

---

**Algorithm 1** Calibrated Normalizing Flows

---

**Input:** Nominal data $\mathcal{D}_0$, target data $\mathcal{D}_t$, step size $\gamma$, number of target subsamples $K$, self-regularization $\beta$
**Output:** Model parameters $\phi$ and calibrated label $c^*$
**for** $k = 1, \ldots, K$ **do**
    $\mathcal{D}_t^{(k)} \leftarrow N$-element random subsample of $\mathcal{D}_t$
**end for**
Initialize $\phi$, $c^*$
**while** $\phi$ not converged **do**
    Update model $\phi \leftarrow \phi - \gamma \nabla_\phi L(\phi, c^*)$
    Update calibration $c^* \leftarrow c^* - \gamma \nabla_c L_{\text{cal}}(\phi, c^*)$
**end while**

---

## 3.1 THEORETICAL ANALYSIS

The secondary optimization of $c^*$ is one of the main differences between CALNF and traditional bootstrapped ensembles (which combine the target candidates in a mixture model rather than optimizing in a lower-dimensional label space). In this section, we provide theoretical motivation for this decision, showing that learning a mapping from labels $c$ to candidate distributions *implicitly* regularizes the learned target posterior.

In particular, consider the Wasserstein metric $W_2(p_1, p_2) = \inf_\gamma \left[ \mathbb{E}_{z_1, z_2 \sim \gamma} ||z_1 - z_2||^2 \right]^{1/2}$ where $\gamma$ is a coupling of probability distributions $p_1$ and $p_2$ The result show that CALNF, in addition to explicitly regularizing the divergence between target candidates, also provides implicit regularization of the $W_2$ metric between the learned nominal and calibrated target posteriors.

**Theorem 1.** *If the flow map $f_\phi(z, c)$ is L-Lipschitz in the second argument, then the Wasserstein distance between the nominal and target posteriors is bounded; $W_2\left( q_\phi(z, \mathbf{0}_K), q_\phi(z, c^*) \right) \leq L||c^*||$*

A proof is included in the appendix, along with $L$ for common normalizing flow architectures.

## 4 EXPERIMENTS

### 4.1 BENCHMARK PROBLEMS

This section describes the benchmark problems used in our experiments; the first is newly developed for our study, but the rest are previously-published benchmarks (Keipour et al., 2021; Deng et al., 2022). We also include the toy 2D problem in Fig. 1 ($N_0 = 10^3$, $N_t = 20$). More details are provided in the appendix, and we provide open-source code and data.

**Air traffic disruptions** We develop a stochastic queuing model of the Southwest Airlines network using publicly available arrival and departure data, based on (Pyrgiotis et al., 2013). The latent variables represent travel times, runway delays, and overnight aircraft reserves. The context includes flight schedules, and observations include actual departure and arrival times. Nominal and failure data from between Dec. 1 through Dec. 20 and Dec. 21 through Dec. 30, respectively, are used for the four busiest airports. The four-airport sub-network has 24 latent variables. We train using $N_0 = 9$ and $N_t = 4$ and evaluate on 4 held-out failure data points (each data point is a single day with between 88–102 flights).

**Aerial vehicle control** We consider a failure detection benchmark for unmanned aerial vehicles (UAVs) using the ALFA dataset (Keipour et al., 2021). This dataset includes real-world data from a UAV during normal flight and during failures with various deactivated control surfaces. The latent variable $z$ parameterizes the nonlinear attitude dynamics and has 21 dimensions, $y$ includes the current and commanded states, and $x$ is the next state. We train on 10 nominal trajectories ($N_0 = 2235$) and 1 failure trajectory ($N_t = 58$) and evaluate on a held-out failure trajectory (69 points).

**Geophysical imaging** Seismic waveform inversion (SWI) is a well-known geophysics problem used as a benchmark for inference and physics-informed learning (Gouveia & Scales, 1998; Deng

Table 1: ELBO (nats/dim) on held-out anomaly data on benchmark problems. 2D and SWI use unseen synthetic data for the test set; all other cases withhold half of the target data for testing. Mean and standard deviation across four seeds are reported. †scaled by $\times 10^{-3}$

|  | 2D
nats/dim ↑ | SWI
nats/dim† ↑ | UAV
nats/dim ↑ | ATC
nats/dim† ↑ |
|---|---|---|---|---|
| KL-regularized ($\beta = 0.01$) | $-3.22_{\pm 0.13}$ | $44.7_{\pm 0.58}$ | $2.87_{\pm 1.42}$ | $-2.26_{\pm 0.10}$ |
| KL-regularized ($\beta = 0.1$) | $-2.03_{\pm 0.04}$ | $44.7_{\pm 0.39}$ | $3.26_{\pm 1.54}$ | $-2.23_{\pm 0.09}$ |
| KL-regularized ($\beta = 1.0$) | $-1.04_{\pm 0.06}$ | $44.3_{\pm 0.40}$ | $3.02_{\pm 1.19}$ | $-2.23_{\pm 0.10}$ |
| $W_2$-regularized ($\beta = 0.01$) | $-4.58_{\pm 0.18}$ | $36.7_{\pm 3.03}$ | $-1.75_{\pm 4.53}$ | $-5.90_{\pm 2.54}$ |
| $W_2$-regularized ($\beta = 0.1$) | $-2.95_{\pm 0.14}$ | $36.7_{\pm 3.02}$ | $-1.54_{\pm 4.31}$ | $-5.80_{\pm 2.45}$ |
| $W_2$-regularized ($\beta = 1.0$) | $-1.67_{\pm 0.05}$ | $36.7_{\pm 2.94}$ | $-2.13_{\pm 5.79}$ | $-6.57_{\pm 4.09}$ |
| Ensemble | $\mathbf{-0.84}_{\pm 0.14}$ | $46.1_{\pm 0.42}$ | $6.65_{\pm 0.98}$ | $-2.23_{\pm 0.06}$ |
| CalNF (ours) | $-0.90_{\pm 0.10}$ | $\mathbf{46.4}_{\pm 0.26}$ | $\mathbf{7.55}_{\pm 0.60}$ | $\mathbf{-2.11}_{\pm 0.13}$ |

et al., 2022; Zhang et al., 2016). SWI seeks to infer the properties of the Earth's subsurface using seismic measurements, which are simulated using the elastic wave equation. This model uses latent variables $z$ for subsurface density, context $y$ for the source signal, and observations $x$ for the seismic measurements (Richardson, 2023). The latent space has 100 dimensions (a $10 \times 10$ grid). We train on $N_0 = 100$ and $N_t = 4$ and evaluate on 500 synthetic samples.

## 4.2 BASELINES AND METRICS

Our main claim is that our CALNF framework is an effective way to learn the posterior when a small number of target data points are available. The most relevant comparisons are to the prior regularization and bootstrapping ensemble methods discussed in Section 2. In particular, we compare against three baselines from previously-published literature: prior regularization using KL divergence (based on Asim et al. (2020)), prior regularization using Wasserstein divergence (based on Finlay et al. (2020) and Onken et al. (2021)), and an ensemble method based on a mixture of normalizing flows (adapted from the generative ensembles proposed in Choi et al. (2019)). The two prior regularization baselines are sensitive to the hyperparameter $\beta$, so we report results for a range $\beta \in [0.01, 1.0]$. CALNF uses $K = 5$ and $\beta = 1.0$ for all problems, and Figs. 11 and 12 in the appendix shows the sensitivity of our method to varying $K$ and $\beta$. We use $K = 5$ components for the ensemble baseline.

Since the large amount of nominal data makes it easy to fit the nominal distribution, we compare primarily on the basis of the evidence lower bound $\mathcal{L}$ computed on held-out target data, reporting the mean and standard deviation over four random seeds. When useful, we also provide visual comparisons of the posterior distributions learned using different methods.

## 4.3 RESULTS & DISCUSSION

Our main empirical results are shown in Table 1. Our method achieves better performance on held-out target data than baselines on all problems. Of course, CALNF's improved performance comes at the cost of increased training time, requiring $K$ additional likelihood evaluations per step relative to the KL- and $W_2$-regularized methods (and the same number of evaluations as the ensemble method). To qualitatively understand the difference in performance between these methods, Fig. 3 compares the learned target posteriors on the SWI problem (which lends itself to easy visualization) with the ground truth in 3a. We see that the KL- and $W_2$-regularized and ensemble methods (Fig. 3b-d) do not infer the correct density profile from the target data, while only our method (Fig. 3e) is able to infer the correct shape. Fig. 4 shows CALNF's prediction in the UAV problem on a held-out failure trajectory after training on only one other failure trajectory (and 10 nominal trajectories). These results suggests that our method is able to appropriately balance the information gained from the nominal distribution with the limited number of target data points.

We also provide the results of an ablation study in Table 8 in the supplementary material, comparing the ELBO achieved when we omit the calibration step (using a constant $c$), omit the nominal data, and remove the subsampling step. These results indicate that most of the performance improvement from CALNF is due to training on random subsamples of the target data. We observe that in cases

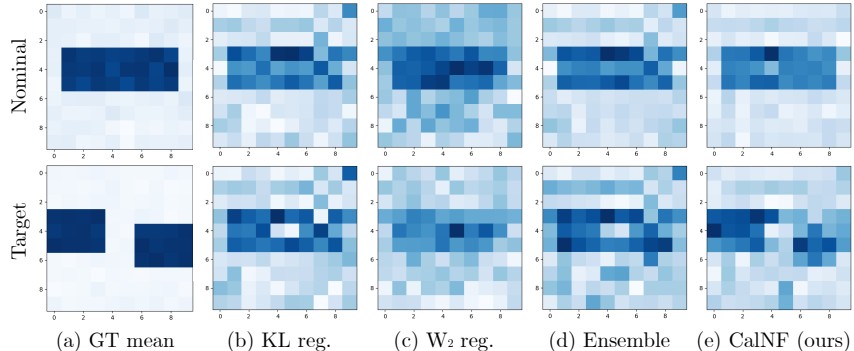

| (a) GT mean | (b) KL reg. | (c) $W_2$ reg. | (d) Ensemble | (e) CalNF (ours) |

Figure 3: **Seismic waveform inversion.** (a) The ground truth nominal and target density profiles. (b-d) The posteriors fit using KL and $W_2$ regularization and CALNF (ours is the only method able to correctly infer the target density profile).

with plentiful nominal data (like the UAV problem), training on zero-labeled nominal data also substantially boosts performance. The appendix includes further results on training time and the sensitivity of CALNF to varying $K$ (Fig. 11), $\beta$ (Fig. 12), and target dataset size $N_t$ (Fig. 12).

## 4.4 USING CALNF FOR ANOMALY DETECTION

Although our main focus in this paper is modeling failure events, we can apply the posteriors learned using our method to detect previously-unseen failures. To use CALNF for anomaly detection, we train the normalizing flow and calibration label as described above, then use the ELBO $\mathcal{L}(\phi, c^*, \{x, y\})$ as the score function to classify a novel datapoint $\{x, y\}$. Table 2 shows the area under the receiver operating characteristic curve (AUROC) and area under the precision-recall curve (AUCPR) for this CALNF-based anomaly detector on the UAV and SWI problems, compared with the supervised anomaly detector proposed in (Gudovskiy et al., 2022; Kang et al., 2022; Rudolph et al., 2021) and with hand-tuned KL- and $W_2$-regularized variants. We were not able to test the ATC problem in this setting due to a lack of data, and the 2D toy example is too simple to be informative, as all methods achieve near-perfect classification. Although CALNF is designed for posterior learning rather than anomaly detection, we find that it achieves more consistent results on this downstream task than existing methods, likely due to CALNF's resistance to overfitting.

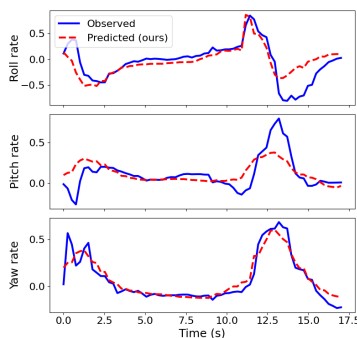

Figure 4: Single-shot UAV failure dynamics predicted by CALNF.

Table 2: Anomaly detection performance on held-out data, reporting mean and standard deviation across four seeds.

|  | AUROC ↑ | | AUCPR ↑ | |
|---|---|---|---|---|
|  | SWI | UAV | SWI | UAV |
| NF-AD | $0.74_{\pm 0.03}$ | $\mathbf{0.65}_{\pm 0.17}$ | $0.77_{\pm 0.03}$ | $0.56_{\pm 0.23}$ |
| NF-AD$_{KL}$ | $0.74_{\pm 0.03}$ | $\mathbf{0.74}_{\pm 0.09}$ | $0.77_{\pm 0.03}$ | $\mathbf{0.70}_{\pm 0.12}$ |
| NF-AD$_{W_2}$ | $0.65_{\pm 0.03}$ | $0.54_{\pm 0.08}$ | $0.63_{\pm 0.03}$ | $0.41_{\pm 0.08}$ |
| Ensemble | $0.75_{\pm 0.10}$ | $0.5_{\pm 0.0}$ | $\mathbf{0.78}_{\pm 0.09}$ | $0.36_{\pm 0.0}$ |
| CalNF | $\mathbf{0.79}_{\pm 0.02}$ | $\mathbf{0.70}_{\pm 0.03}$ | $\mathbf{0.83}_{\pm 0.03}$ | $0.66_{\pm 0.04}$ |

## 4.5 CALNF FOR FEW-SHOT INFERENCE ON IMAGE DATA

Although image modeling is not the focus of this work, for completeness we also include results applying CALNF to few-shot image generation. To use CALNF for this task, we replace the underlying normalizing flow with a conditional Glow architecture (a type of normalizing flow specialized for image modeling Kingma & Dhariwal (2018)), using the calibration label as the

conditioning input. We adapt the MNIST (LeCun et al., 2010) and CIFAR-10 (Krizhevsky, 2009) datasets to create a nominal dataset with all training examples from a single class and a target dataset with 64 examples from a second class. In this case, there is no underlying stochastic process and no context variables (i.e. we directly observe the images $x \sim p_\theta(x|z) = \delta(x - z)$), and the empirical ELBO (4) reduces to the average negative log likelihood of the training data. As a result, minimizing the loss (5) learns an approximation of the training image distribution.

Each method is trained on all examples from the nominal class and the limited set examples from the target class, and we report the negative log-likelihood on held-out test data, then we measure the negative log-likelihood on held-out images from the target class to test generalization beyond the limited training examples. Table 3 and Fig. 6 show the results of this experiment. We find that CALNF is able to generalize better than either a standard or ensemble model, as measured by a higher log likelihood on the held-out data

Table 3: Log-likelihood (bits/dim) on held-out images, reporting mean and standard deviation across four seeds. Higher is better.

|  | MNIST | CIFAR-10 |
|---|---|---|
| Glow (vanilla) | $-5.77_{\pm 0.17}$ | $148.6_{\pm 45.9}$ |
| Glow (ensemble) | $-5.77_{\pm 0.17}$ | $85.91_{\pm 17.5}$ |
| Glow + CALNF | $\mathbf{-5.99}_{\pm 0.12}$ | $\mathbf{23.30}_{\pm 12.9}$ |

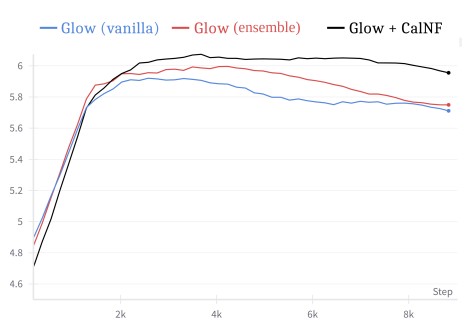

Figure 5: Samples with CALNF (bottom) are more diverse than those without (top).

Figure 6: Log-likelihood (bits/dim) on MNIST validation set; CALNF reduces overfitting to sparse examples.

## 5 CASE STUDY: 2022 SOUTHWEST AIRLINES SCHEDULING CRISIS

In this section, we apply CALNF to a post-mortem analysis of the 2022 Southwest Airlines scheduling crisis. Between December 21st and December 30th, 2022, a series of cascading delays and cancellations severely disrupted the Southwest network, starting in Denver and spreading across the United States. The disruption occurred in roughly two stages, as shown in Fig. 13 in the appendix. In the first stage, from 12/21 to 12/24, weather and operational difficulties caused cancellations to increase from a $< 5\%$ baseline to over $50\%$ of scheduled flights. In the second phase between 12/25 and 12/29, Southwest flight dispatchers started preemptively canceling flights and ferrying crew between airports to reset the network, canceling up to $77\%$ of scheduled flights before returning to near-normal operations on 12/30. Southwest ultimately canceled more than 16,000 flights, affecting more than 2 million passengers, and the airline lost substantial revenue and later paid a \$140 million penalty imposed by the US Department of Transportation ($28\%$ of its 2023 net income; (Rose, 2023)).

This incident has been the subject of extensive investigation, with a report from Southwest Airlines (Southwest Airlines, 2023), testimony before the US Senate from the Southwest Airlines Pilots Association (SWAPA; (Murray, 2023)), and press coverage (Rose, 2023; Cramer & Levenson, 2022). These sources propose a number of hypotheses on the root cause of the 2022 incident. While there is broad agreement that winter weather was a major factor, sources differ on the role of other factors; e.g. the SWAPA report emphasizes poor crew management, while press coverage emphasizes the point-to-point nature of the Southwest network.

Given this context, we have two goals for our case study. First, we are interested in identifying changes in the network state that coincided with the disruption, and how those disrupted parameters compare to the nominal state of the network. Second, we aim to produce a generative model of the disrupted network conditions for use as a tool for network design and analysis (e.g. as a simulation environment for stress-testing future scheduling and recovery policies might).

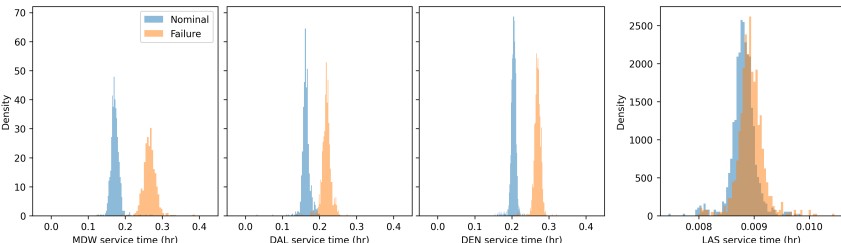

Figure 7: The posterior distribution of departure service times at DEN, MDW, DAL, and LAS. Service times increased only at airports that saw severe weather (DEN, MDW, and DAL).

## 5.1 IMPLEMENTATION

We focus on the first four days of the scheduling crisis, prior to the wave of manual interventions aimed at resetting the network. We conduct our analysis at two spatial resolutions, considering subnetworks of the 4 and 10 busiest airports in the Southwest network, respectively. More details on our model are included in the appendix, along with a key for relevant airport codes in Table 6.

## 5.2 RESULTS

**Localized delays due to winter weather.** Our first observation confirms a common explanation for the disruption: that localized delays at airports across the US coincided with winter weather. Fig. 7 shows CALNF's posterior estimates of nominal and disrupted service times (a proxy for taxi, deicing, and ATC delays) at the four busiest airports. Of these four, only those that experienced severe cold temperatures (DEN, MDW, and DAL) saw an increase in average service time, while there was no significant increase at LAS, which did not have severe weather. This result agrees with official accounts that identify winter weather and a lack of deicing equipment at critical airports like DEN as contributing factors (Southwest Airlines, 2023; Cramer & Levenson, 2022). However, the more important question is how these localized service delays cascaded into the nationwide disruption.

**Cascading failures due to aircraft flow interruption.** Our main finding comes from modeling the movement of aircraft within the network. The number of aircraft starting the day at each airport provides an important measure of robustness, since if there are insufficient aircraft to meet demand, then departing flights must be delayed or canceled. Aircraft deficits can also cascade through the network, as down-stream airports are deprived of the aircraft needed to serve scheduled departures. Because aircraft distribution data are not publicly available, we must use our method to infer it.

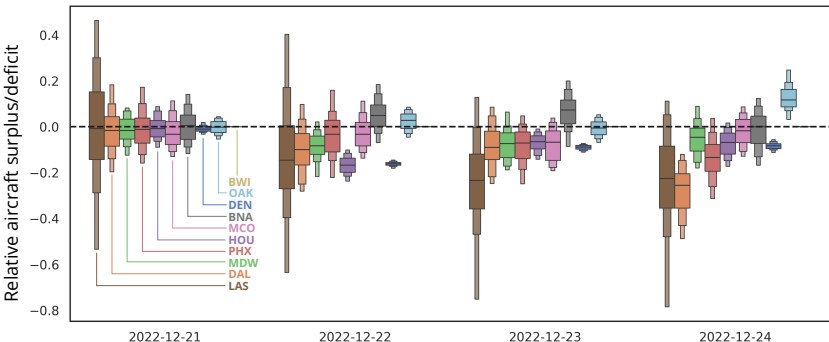

Figure 8: CALNF's inferred posterior estimates of the distribution of Southwest aircraft at the start of the first four days of the disruption, normalized by the number of scheduled departures at each airport; positive/negative indicates more/fewer aircraft than in the nominal case, respectively. CALNF suggests that LAS, DAL, and PHX accumulated a large aircraft deficit over the course of the disruption.

Fig. 8 shows our results from using CALNF to infer the distribution of aircraft reserves in the top-10 network over each of the first four days of the disruption. CALNF finds that there was no detectable deviation from the nominal aircraft distribution on the first day of the disruption, but we infer a steadily increasing deficit at LAS, DAL, and PHX over the following three days. The fact that the aircraft deficit at these airports continued to worsen may have been a factor in Southwest's decision to "hard reset" the network by ferrying empty planes between airports.

Beyond inferring these hidden parameters in the Southwest network, CALNF's results also suggest a possible causal mechanism by which aircraft deficits at LAS, DAL, and PHX propagated to the rest of the network. Although the Southwest network is famously operated in a point-to-point manner, aircraft typically visit multiple airports in sequence on a given day; and disruptions at any of these intermediate destinations can lead to "missing aircraft" at downstream nodes. For example, although LAS and PHX did not experience severe weather during this period, nearly 50% of aircraft ultimately bound for LAS or PHX pass through either DEN or MDW (which did see weather-related delays). Our analysis in Fig. 8 suggests that trends in aircraft reserves at key airports like LAS, PHX, and DAL might be valuable early warning signs for detecting future disruptions.

**Generative modeling**  Once we have learned the nominal and disruption posteriors for the Southwest network, we can use these as generative models for stress-testing proposed modifications to the Southwest network or scheduling system. In future work, we hope to explore how these generative models can be used to design more resilient schedule recovery algorithms.

## 6  CONCLUSION

In this paper, we propose a novel algorithm for rare-event modeling, developing a data-constrained posterior inference tool that uses a subsampling and calibration strategy to avoid overfitting to sparse data. We apply our algorithm to failure analysis and inverse problems, achieving competitive performance on a range of benchmarks with both simulated and real data. We also apply our algorithm to a real-world failure modeling problem, providing new insight into the factors behind the 2022 Southwest Airlines scheduling crisis.

**Limitations & future work**  The primary limitation of our work is that training CALNF on randomly sampled subsets of the target dataset incurs an additional training cost (as shown in Table 9 in the appendix). Although there is no inference-time penalty, CALNF requires $K + 1$ evaluations of the joint likelihood $p(x, z; y)$ per training step (one for each of $K$ subsamples and once for updating $c^*$), compared to $K$ evaluations for the ensemble and one evaluation for the KL- and $W_2$-regularized methods. Our method also requires one additional hyperparameter than the baselines, which adds to implementation complexity, and it is more difficult to train in parallel than the ensemble model. However, even though CALNF is slower to train, it achieves results in the low-data regime that are not possible using faster methods. For example, Fig. 3 shows how our method solves a seismic imaging problem that none of the competing baselines can solve, and CALNF yields better, more consistent performance on downstream anomaly classification tasks. We would also like to emphasize that there is no additional cost for for our method at inference time, other than the negligible cost of passing the calibrated label $c^*$ as an additional input to the normalizing flow.

A second limitation is that our method does not provide an estimate of the risk of failure. Estimating the probability of failure is challenging due to the size of the dataset, but we hope that future work will close this gap; e.g. providing theoretical bounds using large deviation theory (Dembo & Zeitouni, 2010). A final limitation is that we implicitly assume that failure examples share some structure with nominal ones, and so learning a shared representation for both cases (as CALNF does) is helpful. If the failure examples are drawn from a radically different distribution than nominal data, the implicit regularization from this shared representation (discussed in Theorem 1) may not be useful.

**Broader impact**  This paper aims to provide tools to understand the causes of past failures and prevent future incidents. We hope that our work will help enable a more comprehensive data-driven approach to safety analysis for complex systems, including cyberphysical systems and complex infrastructural networks. There is some potential for negative impact (e.g. a bad actor attempting to infer the properties of a safety-critical system to prepare an adversarial attack), but we believe that the potential benefits for designing more robust systems outweigh these concerns.

REPRODUCIBILITY STATEMENT

We include source code for all experiments in a zip file in our supplementary materials, including `README` files with instructions for installing all required dependencies and scripts for recreating all experimental results reported in this paper, including all hyperparameters and random seeds used.

We also provide an open-source implementation of our algorithm and air traffic simulator at `https://github.com/dawsonc/BayesAir`.

ACKNOWLEDGEMENTS

This work was partly supported by the National Aeronautics and Space Administration (NASA) ULI grant 80NSSC22M0070, Air Force Office of Scientific Research (AFOSR) grant FA9550-23-1-0099, and the MIT-DSTA program. Any opinions, findings, and conclusions or recommendations expressed in this publication are those of the authors and do not necessarily reflect the views of the sponsors.

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

## A  LIPSCHITZ CONSTANTS FOR CONDITIONAL NORMALIZING FLOWS

In this section, we provide the Lipschitz constants for various conditional normalizing flow architectures; i.e. $L$ such that $|f(z, c_1) - f(z, c_2)| \leq L \|c_1 - c_2\|$ for all $z, c_1, c_2$.

*Remark* 1. The conditional inverse autoregressive flow (IAF; (Kingma et al., 2016)) has Lipschitz constant $L \leq \prod_i \prod_t (L_{s_t} + L_{m_t})$, where the outer product is over autoregressive blocks and the inner product is over steps within each autoregressive block. $L_{s_t}$ and $L_{s_t}$ are the Lipschitz constants of the neural networks yielding the $m_t$ and $s_t$ values for each autoregressive step (these can be easily bounded for most neural networks; e.g. by the product of the $L_2$ matrix norms of the weight matrices; (Miyato et al., 2018)).

*Remark* 2. A neural spline flow (Durkan et al., 2019) has Lipschitz constant $L \leq 2\bar{s}$, where $\bar{s}$ is an upper bound on the slope $s = (y_{k+1} - y)/(x_{k+1} - x_k)$ between adjacent knot points of the spline (this can be constrained by construction by ensuring a minimum spline bin width).

*Remark* 3. Continuous normalizing flows (Chen et al., 2018) have Lipschitz constant $L \leq e^{L_g \Delta t}$ where $\Delta t$ is the duration of integration and $L_g$ is the Lipschitz constant of the neural network defining the vector field of the flow.

*Remark* 4 (from (Verine et al., 2023)). Normalizing flows based on invertible residual networks, such as i-ResNet (Behrmann et al., 2019) and Residual Flow (Chen et al., 2019), have Lipschitz constant $L \leq (1 + L_g)^m$, where $m$ is the number of residual blocks and $L_g < 1$ is the Lipschitz constant of the residual block $g(x)$.

*Remark* 5 (from (Verine et al., 2023)). Normalizing flows based on Glow (Kingma & Dhariwal, 2018) have Lipschitz constant $L \leq \prod \|W_i\|_2$, where the product is over the weight matrices $W_i$ of the convolution blocks.

## B    PROOF OF LEMMA 1

*Proof.* The optimal maximum likelihood estimator is a mixture of delta functions $q^*(z) = \sum_{z^{(i)} \in \mathcal{D}} \delta(z - z^{(i)})/N$. Since we assume that the data points are well-separated, the optimal $L$-Lipschitz maximum likelihood estimator replaces the scaled delta $\delta(z)/N$ with the $L$-Lipschitz function $\hat{\delta}(z)$ that a) is non-negative, b) integrates to $1/N$, and c) maximizes the value $\hat{\delta}(0)$. Constraint (c) will be active (otherwise we recover the scaled delta function), so we know that $\hat{\delta}$ will have the form $max(0, a - L\|z\|)$. Normalizing to $1/N$ yields

$$\hat{\delta}(z) = \max\left(0, a - L|z|\right) \tag{6}$$

$$a = \left(\frac{(d+1)L^d\Gamma(\frac{d}{2}+1)}{\pi^{d/2}N}\right)^{1/(d+1)} \tag{7}$$

Substituting one data point for another (subject to the assumption on data sparsity) changes the optimal estimator by swapping the corresponding mixture component; i.e. exchanging $\hat{\delta}(z - z^{(2)})$ for $\hat{\delta}(z - z^{(1)})$. The $W_2$ distance between the resulting mixtures is the same as the $W_2$ distance between the changed components. Each component has probability mass $1/N$ and they are distance $\|z^{(1)} - z^{(2)}\|$ apart, completing the proof. $\qquad\square$

## C    PROOF OF LEMMA 2

*Proof.* Let $\mathcal{D}_i$ be a random dataset created by sampling $\mathcal{D}$ $N$ times with replacement. In the proof of Lemma 1, we show that the optimal $L$-Lipschitz maximum likelihood estimator given $\mathcal{D}_i$ is $\sum_{z \in \mathcal{D}_i} \hat{\delta}(z - z^{(i)})$. This gives ensemble model

$$q_{\text{ensemble}}(z) = \sum_{i=1}^{K} \left[\sum_{z \in \mathcal{D}_i} \hat{\delta}(z - z^{(i)})\right] / K \tag{8}$$

Since each $\mathcal{D}_i$ is sampled independently with replacement, we can combine the nested sums

$$q_{\text{ensemble}}(z) = \sum_{z^{(i)} \in \mathcal{D}_{NK}} \hat{\delta}(z - z^{(i)})/K \tag{9}$$

where $\mathcal{D}_{NK}$ is a single dataset of $NK$ points sampled with replacement from $\mathcal{D}$. As $K \to \infty$, the empirical distribution of $\mathcal{D}_{NK}$ approaches that of $\mathcal{D}$, so this sum reduces almost surely to:

$$\lim_{K \to \infty} q_{\text{ensemble}}(z) \overset{a.s.}{=} \sum_{z^{(i)} \in \mathcal{D}} \hat{\delta}(z - z^{(i)}) \tag{10}$$

$\qquad\square$

## D  PROOF OF THEOREM 1

*Proof.* The $W_2$ metric is defined as an infemum over couplings $\gamma$, so in order to provide an upper bound it suffices to propose a coupling between the nominal and target posteriors, $q_\phi(z, \mathbf{0}_K)$ and $q_\phi(z, c^*)$. Recall that the normalizing flow $q_\phi$ has base distribution $q_0$ and flow map $f_\phi$, where $f_\phi(z, c)$ is assumed to be $L$-Lipschitz in the second argument. Consider the joint distribution $\gamma(z_1, z_2)$ defined by $z_0 \sim q_0(z)$, $z_1 = f_\phi(z_0, \mathbf{0}_K)$, and $z_2 = f_\phi(z_0, c^*)$. By construction, the marginals of $\gamma$ in each argument are $q_\phi(z, \mathbf{0}_K)$ and $q_\phi(z, c^*)$, respectively, and so $\gamma$ is a valid coupling.

This provides the bound

$$
\begin{aligned}
W_2(q_\phi(\cdot, \mathbf{0}_K), q_\phi(\cdot, c^*)) &\leq \left[ \mathop{\mathbb{E}}_{z_1, z_2 \sim \gamma} ||z_1 - z_2||^2 \right]^{1/2} \\
&\leq \left[ L^2 ||c^* - \mathbf{0}_K||^2 \right]^{1/2} \\
&\leq L ||c^*||
\end{aligned}
$$

$\square$

## E  DETAILS ON BENCHMARK PROBLEMS

This section provides additional details for the three types of inverse problem studied in our paper. All problems are implemented using the Pyro probabilistic programming framework (Bingham et al., 2019).

### E.1  SEISMIC WAVEFORM INVERSION

An illustration of the SWI problem is given in Fig. 9. We implement the SWI problem using the Deepwave library (Richardson, 2023). We use latent parameters $z \in \mathbb{R}^{n_x \times n_y}$ representing the subsurface density profile (with spatial resolution $n_x = 10$ and $n_y = 10$), context $y \in \mathbb{R}^{n_T}$ representing the source signal, and observations $x \in \mathbb{R}^{n_s \times n_r \times n_T}$ representing the signal measured at each receiver, where $n_s = 1, n_r = 9, n_T = 100$ are the number of sources, receivers, and timesteps, respectively. Before solving the elastic wave PDE, the density profile is upsampled to $100 \times 30$. The observations are corrupted with additive isotropic Gaussian noise. The parameters of this problem are summarized in Table 4.

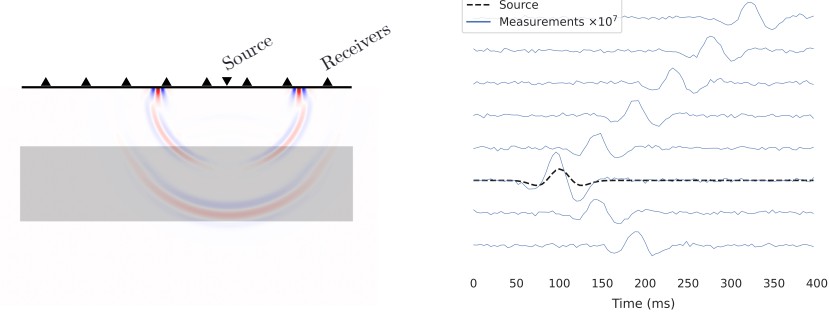

Figure 9: (Left) An illustration of the SWI problem and (right) the receiver measurements (blue) given a source signal (black).

### E.2  UAV CONTROL

We model the nonlinear attitude dynamics of the UAV as a combination of an unknown linear mapping from the current and desired states to angular rates, then a nonlinear mapping from angular rates to updated UAV orientation. The state $q = [\phi, \theta, \psi]$ includes the roll, pitch, and yaw angles of the UAV,

Table 4: Summary of parameters for the SWI problem.

|  | Dimension |
|---|---|
| **Latent parameters $z$** | |
| Density profile ($10 \times 10$) | 100 |
| **Context $y$** | |
| — | |
| **Observation $x$** | |
| Seismic waveform (100 timesteps at 9 receivers) | 900 |

and $\hat{q}$ denotes the commanded orientation. We model the angular rates of the UAV as

$$\omega = \begin{bmatrix} p \\ q \\ r \end{bmatrix} = Aq + K(\hat{q} - q) + d + \eta \tag{11}$$

where $A$, $K$, and $d$ are unknown feedforward, feedback, and bias dynamics, and $\eta$ is Gaussian process noise. The state derivative is related to $\omega$ by

$$\frac{d}{dt}q = J^{-1}(q)\omega \tag{12}$$

$$J^{-1}(q) = \begin{bmatrix} 1 & \tan(\theta)\sin(\phi) & \tan(\phi)\cos(\theta) \\ 0 & \cos(\phi) & -\sin(\phi) \\ 0 & \sin(\phi)/\cos(\theta) & \cos(\phi)/\cos(\theta) \end{bmatrix} \tag{13}$$

We apply a first-order time discretization to yield the one-step stochastic dynamics

$$q_{t+1} = q_t + \delta_t J^{-1}(q)\left(Aq + K(\hat{q} - q) + d + \eta\right)$$

and observed states are additionally corrupted by Gaussian noise. A summary of the parameters for this problem are given in Table 5.

An example trajectory, including both nominal and anomalous segments, for the UAV dataset are shown in Fig. 10. In this case, the anomaly is relatively easy to detect; the challenge is understanding how the aircraft's flight dynamics change during the failure so that a recovery controller can be designed to handle this case.

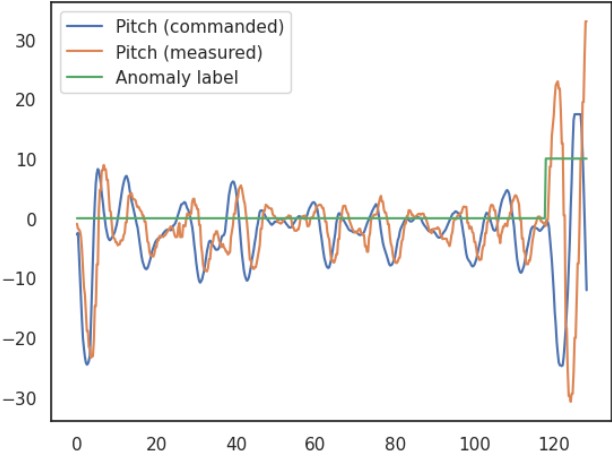

Figure 10: Example trajectory that includes an elevator failure, including both nominal and anomalous segments.

Table 5: Summary of parameters for the UAV problem.

|  | Dimension |
|---|---|
| **Latent parameters** $z$ | |
| Feedforward matrix $A$ ($3 \times 3$) | 9 |
| Feedback matrix $K$ ($3 \times 3$) | 9 |
| Bias term $d$ | 3 |
| **Context** $y$ | |
| Current state | 3 |
| Desired orientation | 3 |
| **Observation** $x$ | |
| Next state | 3 |

### E.3 AIR TRAFFIC NETWORK

The input to our air traffic model is a list of scheduled flights, each specifying an origin and destination airport and a scheduled departure and arrival time. The latent state $z$ includes the mean travel time between each origin/destination pair, the mean service time at each airport (which affects both arriving and departing aircraft and models taxi, deicing, and ATC delays), the mean turnaround time at each airport (the minimum time that must elapse before an arriving aircraft may depart), the baseline cancellation rate at each airport, and the initial number of aircraft at each airport. A summary of these parameters are given in Table 7. So that the benchmarks in Section 4 can be run in a reasonable time, we restrict the ATC problem used for benchmarking to the four busiest airports and do not model cancellations, but we use the ten busiest airports and do include cancellations in our case study in Section 5.

The model steps through the scheduled flights in 15 minute increments. In each increment, it checks for the flights that are scheduled to depart from each airport. Each of these flights receives a certain probability of cancellation given by

$$P(\text{canceled}) = 1 - (1 - p_c)\sigma \left( 10 \frac{\text{\# available aircraft}}{\text{\# departing flights in this block}} \right) \tag{14}$$

where $p_c$ is the baseline cancellation rate for the origin airport and $\sigma$ is the sigmoid function, so the probability of cancellation is $p_c$ when there are more available aircraft than scheduled departures and approaches 1 as the number of available aircraft decreases. Cancellations are sampled from a relaxed Bernoulli distribution with this cancellation probability and a straight-through gradient estimator. If a flight is canceled, it is marked as such and the observation for that flight will just be `canceled` and will not include actual departure and arrival times. If the flight is not canceled, then it is moved to the runway queue if there are enough aircraft available; otherwise, it is delayed until the next time block.

Both departing and arriving flights are served using a single M/M/1 queue for each airport, with service times drawn from an exponential distribution with the mean specified according to each airport's mean service time. Once airborne, departing flights are assigned a random flight time from a Gaussian with mean given by the mean travel time for each route and fixed variance. Once this travel time has elapsed, they enter the runway queue at the destination airport. Once an aircraft has landed, it does not become available to serve new flights until the minimum turnaround time has elapsed (which is sampled from a Gaussian with mean given by the mean turnaround time for each airport). Observations for non-canceled flights include the simulated arrival and departure times, plus some fixed-variance Gaussian noise.

### E.4 TOY 2D PROBLEM

The data for the 2D toy problem is generated by uniformly sampling nominal data:

$$\theta \sim \mathcal{U}(0, \pi)$$
$$x \sim \mathcal{N}(\cos\theta - 0.5, 0.1)$$
$$y \sim \mathcal{N}(\sin\theta - 0.25, 0.1)$$

Table 6: International Air Transport Association (IATA) codes and full names of the ten busiest airports in the Southwest network.

| | |
|---|---|
| DEN | Denver International Airport |
| DAL | Dallas Love Field Airport |
| MDW | Chicago Midway International Airport |
| PHX | Phoenix Sky Harbor International Airport |
| HOU | William P. Hobby Airport |
| LAS | McCarran International Airport |
| MCO | Orlando International Airport |
| BNA | Nashville International Airport |
| BWI | Baltimore/Washington International Thurgood Marshall Airport |
| OAK | Oakland International Airport |

Table 7: Summary of parameters for the ATC problem. $n_{airport}$ indicates the number of airports in the model. $n_{flights}$ indicates the total number of scheduled flights. $^{\dagger}$ indicates parameters that are only included in the case study.

| | Dimension | Top-4 (Section 4) | Top-10 (Section 5) |
|---|---|---|---|
| **Latent parameters** $z$ | | | |
| Log. of turnaround time at each airport (mean minimum delay between arrival and departure) | $n_{airport}$ | 4 | 10 |
| Log. of service time at each airport (mean delay between pushback and takeoff) | $n_{airport}$ | 4 | 10 |
| Log. of mean travel times between each airport | $n_{airport}^2$ | 16 | 100 |
| Log. of initial aircraft reserves at each airport | $n_{airport}^{\dagger}$ | – | 10 |
| Log. of baseline cancellation probability at each airport | $n_{airport}^{\dagger}$ | – | 10 |
| **Context** $y$ | | | |
| Scheduled arrival time of each flight | $n_{flights}$ | 44–102 | 405–497 |
| Scheduled departure time of each flight | $n_{flights}$ | 44–102 | 405–497 |
| **Observation** $x$ | | | |
| Actual arrival time of each flight | $n_{flights}$ | 44–102 | 405–497 |
| Actual departure time of each flight | $n_{flights}$ | 44–102 | 405–497 |
| Whether each flight was cancelled | $n_{flights}$ | 44–102 | 405–497 |

and anomaly data

$$\theta \sim \mathcal{U}(\pi, 2\pi)$$
$$x \sim \mathcal{N}(\cos\theta + 0.5, 0.1)$$
$$y \sim \mathcal{N}(\sin\theta + 0.75, 0.1)$$

Since this problem is meant as an easy-to-visualize test for whether a method can learn a posterior distribution with a complex shape, we set $[x, y]$ as the latent parameters and assume they are observed directly (with the addition of Gaussian noise), rather than treating $\theta$ as the latent parameter (which would lead to a very easy-to-fit posterior).

## F IMPLEMENTATION DETAILS

We implement CALNF using neural spline flows (NSF) as the underlying normalizing flow (Durkan et al., 2019). We note that CALNF is agnostic to the underlying flow architecture; we also tried using masked autoregressive flows (Huang et al., 2018), which trained faster but had slightly worse performance, and continuous normalizing flows (Chen et al., 2018), which trained much more slowly.

We implement the KL regularization baseline using neural spline flows with a KL regularization penalty between the learned anomaly and nominal posteriors. We implement an RNODE-derived method that includes only the $W_2$ regularization term, not the Froebenius norm regularization term

(which is used only to speed training and inference, not to regularize the learned posterior; (Finlay et al., 2020)).

We extend our method to anomaly detection by defining a score function as the ELBO of a given observation, approximated using 10 samples from the learned posterior.

All methods were implemented in Pytorch using the Zuko library for normalizing flows (Zuk, 2024). The neural spline flows used 3 stacked transforms, and all flows used two hidden layers of 64 units each with ReLU activation (except for the continuous flows on the 2D problem, which use two hidden layers of 128 units each). All flows were trained using the Adam optimizer with the learning rate $10^{-3}$ (except on the UAV problem, which used a learning rate of $10^{-2}$) and gradient clipping. CALNF used $K = 5$ on all problems. All methods were trained on either a single NVIDIA GeForce RTX 2080 Ti GPU or a `g4dn.xlarge` AWS instance, with 200, 500, 500, and 150 epochs for the 2D, SWI, UAV, and ATC problems, respectively. The image generation examples were trained for 1000 epochs (MNIST) and 500 epochs (CIFAR-10). Non-image benchmarks all take less than 2 hours to train, and image benchmarks take approximately 20 hours to train. We estimate cloud compute costs for this entire project (including preliminary experiments) at less than 200 USD.

Code examples, including scripts for reproducing the results in Tables 1 and 8 and notebooks containing our data analysis for Section 5, are included in the attached supplementary material. An open-source version of our algorithm and air traffic network simulator is available at `https://github.com/dawsonc/BayesAir`.

## G  FURTHER EMPIRICAL RESULTS

Table 8: ELBO (nats/dim) on held-out target data for ablations of CALNF. The first is our proposed method, the second fixes $c$, the third excludes the nominal data during training, and the fourth does not subsample the target data. [†]scaled by $\times 10^{-3}$

|  | 2D | SWI[†] | UAV | ATC[†] |
|---|---|---|---|---|
| CALNF | $-\mathbf{0.90}_{\pm 0.1}$ | $\mathbf{46.3}_{\pm 0.2}$ | $6.95_{\pm 1.2}$ | $-\mathbf{2.01}_{\pm 0.1}$ |
| w/o $c^*$ | $-0.96_{\pm 0.2}$ | $46.2_{\pm 0.4}$ | $\mathbf{7.86}_{\pm 1.0}$ | $-2.02_{\pm 0.1}$ |
| w/o $\mathcal{D}_0$ | $-1.12_{\pm 0.2}$ | $46.1_{\pm 0.4}$ | $-9.22_{\pm 10}$ | $-2.03_{\pm 0.2}$ |
| w/o $\mathcal{D}_t^i$ | $-1.03_{\pm 0.2}$ | $43.9_{\pm 2.8}$ | $-3.65_{\pm 11}$ | $-2.05_{\pm 0.1}$ |

Fig. 11 shows the sensitivity of our method to different values of $K$ on the SWI benchmark. We find that there is a slight trend towards better performance as $K$ increases, and that including the calibration step (rather than using a fixed $c^*$) improves performance at all levels of $K$.

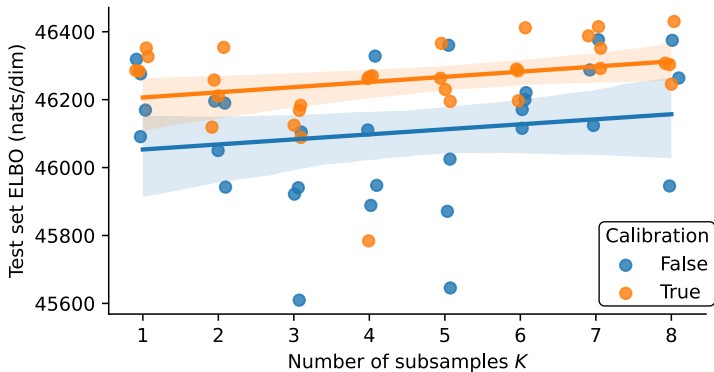

Figure 11: The ELBO on a held-out test set for the anomaly posterior learned using CALNF on the SWI example using a varying number of subsamples.

Table 9 includes the training times for our method and each baseline. We can also compare these methods theoretically in terms of the number of evaluations of the joint likelihood $p(x, z; y)$ in Eq. (4).

Table 9: Training time (minutes) on benchmark problems. Mean and standard deviation reported over four random seeds.

|  | 2D | SWI | UAV | ATC |
|---|---|---|---|---|
| KL-regularized ($\beta = 0.01$) | $1.25_{\pm 0.30}$ | $6.48_{\pm 0.14}$ | $31.17_{\pm 0.37}$ | $84.29_{\pm 1.32}$ |
| KL-regularized ($\beta = 0.1$) | $1.26_{\pm 0.13}$ | $6.53_{\pm 0.18}$ | $31.20_{\pm 0.34}$ | $84.39_{\pm 0.88}$ |
| KL-regularized ($\beta = 1.0$) | $1.34_{\pm 0.05}$ | $6.58_{\pm 0.18}$ | $31.24_{\pm 0.16}$ | $84.09_{\pm 0.78}$ |
| $W_2$-regularized ($\beta = 0.01$) | $8.70_{\pm 0.43}$ | $8.50_{\pm 0.25}$ | $23.93_{\pm 0.87}$ | $94.64_{\pm 1.36}$ |
| $W_2$-regularized ($\beta = 0.1$) | $8.59_{\pm 0.49}$ | $8.27_{\pm 0.42}$ | $23.54_{\pm 0.62}$ | $93.98_{\pm 1.58}$ |
| $W_2$-regularized ($\beta = 1.0$) | $8.40_{\pm 0.50}$ | $8.40_{\pm 0.38}$ | $23.96_{\pm 0.54}$ | $93.99_{\pm 1.22}$ |
| Ensemble | $1.58_{\pm 0.26}$ | $14.25_{\pm 0.30}$ | $76.24_{\pm 0.28}$ | $147.91_{\pm 6.81}$ |
| CalNF (ours) | $2.35_{\pm 0.37}$ | $19.74_{\pm 0.28}$ | $77.74_{\pm 0.71}$ | $174.49_{\pm 6.87}$ |

The KL- and $W_2$-regularized methods require one evaluation per training step, the ensemble method requires $K$ evaluations, and our method requires $K + 1$. The inference times are identical except for the $W_2$-regularized method, which is slower due to its use of neural ODEs for the flow map).

Fig. 12 show the performance of CALNF on the 2D benchmark as the number of target data points is decreased. When the size of the target dataset is at least 10% the size of the nominal dataset, CALNF's test-set ELBO saturates at a high value. Between 2% and 5%, CALNF's performance begins to deteriorate, and below 2% performance drops off sharply. Higher values of $\beta$ improve performance in extremely data-sparse cases, but decrease performance when many target data points are available. This example uses 1000 nominal data points; the results in our main paper use $\beta = 1.0$ (constant across all experiments) and a target dataset 2% of the size of the nominal dataset for this problem.

From the results in Fig. 12, we see that larger values of $\beta$ perform better when the failure dataset is very small, but smaller values of $\beta$ perform better when the failure dataset is large. These results suggest that when plenty of information is available (in the form of a large training dataset), it is beneficial to encourage diversity among the candidate posteriors through a small regularization strength $\beta$; however, when information is limited, we can achieve better performance by encouraging similarity between candidate posteriors to reduce overfitting.

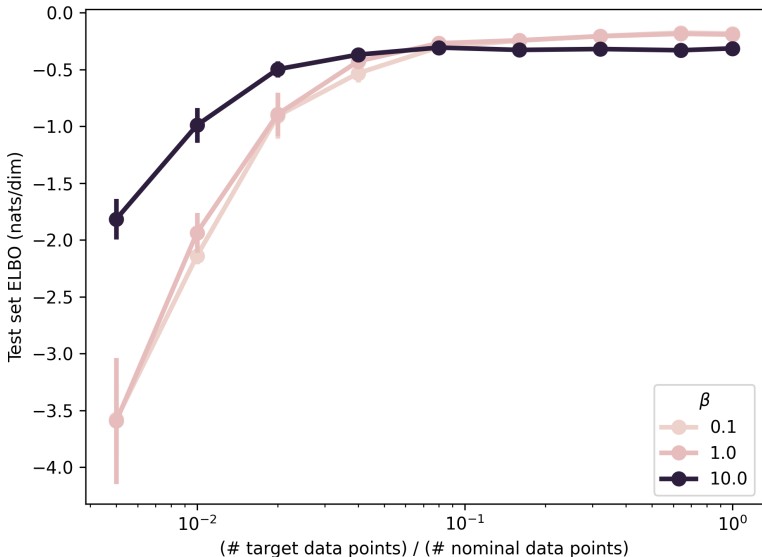

Figure 12: Performance of CALNF under extreme data scarcity on the 2D benchmark, with varying self-regularization strengths $\beta$.

Table 10: ELBO (nats/dim) on held-out anomaly data of the ensemble baseline trained without nominal data. The second and third rows are replicated from Table 1 for ease of comparison.

| | 2D (nats/dim) ↑ | SWI (nats/dim) ↑ | UAV (nats/dim) ↑ | ATC (nats/dim) ↑ |
|---|---|---|---|---|
| Ensemble (w/o nominal data) | $-1.33_{\pm 0.23}$ | $45.93_{\pm 0.61}$ | $-5.81_{\pm 3.41}$ | $-2.07_{\pm 0.11}$ |
| Ensemble (w/ nominal data) | $-0.84_{\pm 0.14}$ | $46.1_{\pm 0.42}$ | $6.65_{\pm 0.98}$ | $-2.23_{\pm 0.06}$ |
| CalNF (ours) | $-0.90_{\pm 0.10}$ | $46.4_{\pm 0.26}$ | $7.55_{\pm 0.60}$ | $-2.11_{\pm 0.13}$ |

Table 10 includes additional results comparing the performance of the ensemble method both with and without simultaneous training on the nominal dataset, with the performance of our method from Table 1 included for ease of reference.

## H  ADDITIONAL RESULTS ON SOUTHWEST AIRLINES CASE STUDY

A timeline of the 2022 Southwest Airlines scheduling crisis is shown in Fig. 13.

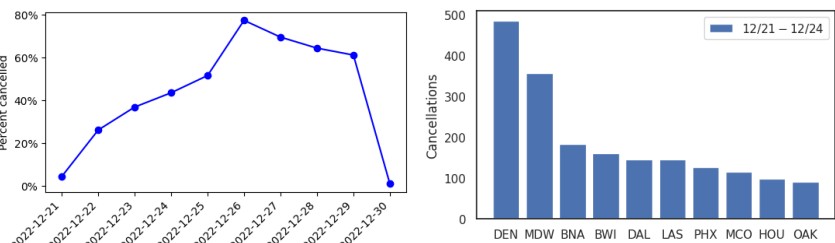

Figure 13: (Left) Timeline of cancellations during the 2022 Southwest Airlines scheduling crisis. (Right) Cancellations at the 10 busiest airports during the first four days of the disruption.

