# OpenReview forum: "Rare event modeling with self-regularized normalizing flows: what can we learn from a single failure?"
_ICLR.cc/2025/Conference — ICLR 2025 Poster_

### Official Review · Reviewer_mQHM · 2024-10-29

**Soundness:** 3
**Presentation:** 3
**Contribution:** 2
**Rating:** 5
**Confidence:** 3

**Summary:**

The paper introduces Calibrated Normalizing Flows (CALNF), a novel approach to rare event modeling in data-constrained scenarios. CALNF builds on normalizing flows to learn an approximate posterior over latent variables $z$, using a blend of ensemble techniques and self-regularization. In essence, the method operates by jointly training on the complete target dataset, its subsets, and nominal data. A conditional normalizing flow is applied across both the target, nominal and subset posteriors, followed by a calibration step that optimally combines these learned posteriors to approximate the overall target distribution. By jointly learning across both subsets and the full dataset, CALNF captures the target distribution effectively while minimizing overfitting, making it more robust in settings with limited data.

While the paper presents an innovative approach to rare event modeling, the marginal performance gains over the ensemble baseline do in my opinion not justify the increased computational complexity, leading to a weak reject. For a more detailed justification, please see the comments below.

**Strengths:**

**Addresses an important problem:** The paper tackles the challenge of modeling rare events with limited data which is a relevant issue in many safety-critical applications.

**Theoretical Foundation:** The paper includes a theoretical analysis, such as bounding the Wasserstein distance between nominal and target posteriors, adding robustness to the method.

**Clear Presentation:** The method and its applications are well-presented, making the paper easy to read.

**Weaknesses:**

**Marginal performance gains vs computational complexity**  As the authors note, Table 1 contains the main empirical results, comparing performance on four benchmark datasets between the proposed method and simpler ablated versions, including the Ensemble approach. However, the Ensemble method performs nearly as well as the proposed CALNF, with results that are statistically indistinguishable from CALNF in three out of four cases. This similarity in performance raises questions about whether the added complexity of CALNF is justified. In particular, (i) the Ensemble approach has fewer hyperparameters (only the number of ensemble components, with no regularization parameter), (ii) does not rely on nominal data, and (iii) allows parallelized training across ensemble members.

While the issue of additional hyperparameters is mitigated to some extent by the authors' ablation study in the appendix, the other limitations are more concerning. CALNF requires joint training on nominal and target data to estimate the posterior over the target distribution, adding significant complexity relative to the Ensemble method. This dependency on both data types also limits CALNF’s suitability for online or real-time applications, where training with nominal data might not be feasible, further reducing its practical applicability.

**Questions:**

I recommend that the authors further clarify and refine the benefits of CALNF compared to the Ensemble method, particularly addressing cases where the performance gains justify the additional complexity.

---

> ### Author Response · Authors · 2024-11-22
>
> Thank you for your feedback. You raise an important point r.e. justifying CalNF’s increased complexity with corresponding increased performance.
>
> First, you are correct that the ensemble approach has fewer hyperparameters (one fewer than our method) and is easier to parallelize. We have added discussion of these points to the limitations section of our revised paper.
>
> Second, we would like to clarify point (ii) in your review: in our implementation, the ensemble model is trained on both nominal and target data (using a label to distinguish between the two, implemented similarly to $c$ in our method). We include a table below (and in the appendix of our revised paper) that shows the results of training the ensemble model on the target data alone. We find that excluding the nominal data decreases the performance of the ensemble model on three out of four problems. In addition, all of the regularization-based methods also require training on nominal as well as target data.
>
> |                             |   2D (nats/dim) 🡑  |  SWI (nats/dim) 🡑  |  UAV (nats/dim) 🡑  |  ATC (nats/dim) 🡑  |
> |-----------------------------|:------------------:|:------------------:|:------------------:|:------------------:|
> | Ensemble (w/o nominal data) | $-1.33_{\pm 0.23}$ | $45.93_{\pm 0.61}$ | $-5.81_{\pm 3.41}$ | $-2.07_{\pm 0.11}$ |
> | Ensemble (w/ nominal data)  | $-0.84_{\pm 0.14}$ |  $46.1_{\pm 0.42}$ |  $6.65_{\pm 0.98}$ | $-2.23_{\pm 0.06}$ |
> | CalNF (ours)                | $-0.90_{\pm 0.10}$ |  $46.4_{\pm 0.26}$ |  $7.55_{\pm 0.60}$ | $-2.11_{\pm 0.13}$ |
>
> Third, while the results in Table 1 shows that our method achieves modest performance gains over the ensemble method on the posterior learning task (as you point out in your review), Table 2 provides important context for how posterior learning affects downstream tasks like anomaly classification. The results in Table 2 show that our method yields better, more consistent performance than all other methods on this downstream anomaly classification task.
>
> Finally, regarding online or real-time applications: our method is designed primarily with post-event failure analysis (i.e. “post-mortem” analysis), but if we were to deploy CalNF in real-time (e.g. for failure detection), we would pre-train on any available data offline, then use CalNF for online inference to classify new examples as nominal or failure. While CalNF is more complex to train than an ensemble model, it is faster at inference time (since we only need to evaluate one model with the optimized label rather than evaluating all mixture members). In this application, we would expect the improved performance of CalNF on downstream tasks like failure detection (in Table 2) to be even more important.
>
> Does this help clarify your questions from your review? We are happy to follow up if you have further questions or if there are any points that you are not satisfied with.

---

> > ### Comment · Reviewer_mQHM · 2024-11-25
> >
> > Thanks for your answer and running the additional experiment. While I appreciate your effort, I will keep the score as it is.
> >
> > My main concern remains the empirical comparison to the Ensemble baseline. While I understand your argument that Table 2 demonstrates improved performance on a downstream task, the paper's focus is centered on modeling failure events rather than anomaly detection. If Table 2 is to play a central role, this would necessitate a rewrite of the storyline and likely require additional experiments in this direction. While this is a valid approach, it is outside of the scope of this rebuttal.

---

> > > ### Author Response · Authors · 2024-11-25
> > >
> > > We appreciate your feedback and your time engaging in this review process.
> > >
> > > If we may clarify: you are right that our paper focuses on modeling failure events (i.e. learning the posterior distribution), but there are some issues with the ELBO metric in Table 1 that make it helpful to look at performance on downstream tasks as well.
> > >
> > > In particular, small changes in ELBO can lead to large changes in qualitative performance. For example, consider the SWI problem, where our method achieves a mean of 46.4 and ensemble achieves 46.1. Despite this relatively small difference in ELBO, when we visualize the learned posterior distributions (Fig. 3) we see that the ensemble method has failed to learn key features that our method picks up on (e.g. the gap between the two shaded regions in the ground truth).
> > >
> > > This sensitivity makes it helpful to look at performance on downstream tasks (anomaly detection in Table 2 and few-shot image learning in Table 3) to supplement the results in Table 1. Even though our method and the ensemble method yield similar test-set ELBO on some problems, the fact that our method performs better on downstream tasks suggests that our method is learning posteriors that are more robust and more useful than those learned by other methods.
> > >
> > > Again, we appreciate your time engaging in this review process and your comments. If you have any additional feedback, we would be happy to hear it.

---

### Official Review · Reviewer_Qbv4 · 2024-10-30

**Soundness:** 3
**Presentation:** 3
**Contribution:** 2
**Rating:** 6
**Confidence:** 4

**Summary:**

In this work, the authors propose a calibrated normalizing flow method to address the rare-event modeling problem in data-constrained settings. Experimental validation demonstrates improved performance compared to the prior regularization and bootstrapping ensemble baselines.

**Strengths:**

The paper is well-written, and the proposed method is presented clearly.

Experimental settings and results are described in detail, and limitations related to training costs are analyzed.

Extensive experimental validations across various settings demonstrate the effectiveness of the proposed method.

**Weaknesses:**

A subsection that clearly outlines a comprehensive literature review of recent research achievements in related work is missing.

In the primary experimental validation, the most relevant comparisons are made between prior regularization techniques (based on studies from 2020/2021) and the bootstrapping approach (from 2019). Thus, claiming that the proposed method achieves state-of-the-art performance in both the abstract and introduction (lines 63–64) seems not very convincing.

In Section 4.2, the ensemble size is set to $K=5$. Additionally, an ablation study examines both the training time and the sensitivity of CalNF to varying values of $K$ (These results are appreciated). As increasing ensemble size generally improves performance, it remains uncertain whether the proposed method maintains its superiority over ensemble baselines (as shown in Table 1) as $K$ increases.

**Questions:**

**Q1:** Equation 5 and the explanation between lines 175-177 appear to present conflicting information. If minimizing the total loss leads to maximizing the KL divergence between candidate posteriors (due to the subtraction sign), wouldn’t this be against the statement that the divergence between candidate posteriors learned for each subsample should be small?

**Q2:** Could you please elaborate on why the candidate posteriors learned for each subsample should be kept small? Typically, ensemble methods enhance predictive performance by increasing the diversity among ensemble members.

**Q3:** Is there a typo in the Equation 3 where the $q_{\phi_t}$ in the DL part should be $q_{\phi}$? Should $c^*$ be in the loss $L_{cal}$ (line 178) rather than $c$?

**Q4:** Could you please elaborate on why the ensemble method can not infer the correct shape (lines 314-316) in Figure 3? The results of the ensemble and the CalNF look a bit similar.

**Q5:** In lines 112-114, some drawbacks of the ensemble method (e.g., data sensitivity issue in deep models), would the proposed method counter these issues?

**Q6:** Regarding the statements in lines 129-131, could you please provide some supportive references or explanations to illustrate the practical issues even in the case of selecting an optimal $\beta$?

---

> ### Author Response · Authors · 2024-11-22
>
> Thank you for your review and feedback! We have included responses to your questions below along with a revised version of our paper (with changes highlighted in blue).
>
> **Weakness 1 (lit review):** Thank you for this feedback. We had discussed prior work in the introduction and background sections of our paper, but we agree that there should be a focused subsection on related work. We have added this subsection in our revised paper.
>
> **Weakness 2 (baselines):** We selected baselines that we believe are representative of prior work on this problem. Most of the work in the last few years on few-shot generative modeling has focused on image generation (see Abdollahzadeh et al., 2023 in our bibliography for a survey). Unfortunately, these methods for few-shot image generation are typically not applicable to non-image domains, since they rely on pre-training on massive open image datasets and it is not clear how they would be applied when large domain-specific datasets are not available. We have included further discussion of this in the revised related work section. If you have suggestions for specific methods for us to consider, we would be happy to discuss those.
>
> **Weakness 3 (increasing ensemble size):** While increasing the ensemble size generally improves performance on problems with plentiful data, this effect is much less strong in data-constrained settings. This is because each ensemble member is trained on a random subsample of the target dataset, and when this dataset is small there are only a limited number of possible distinct subsamples (e.g. only 35 possible combinations for $N_t=4$, as in the ATC problem). Even before $K$ reaches this limit, each new subsample becomes more and more similar to previously-drawn subsamples. This is why our results in Fig. 11 show that the performance of our method increases slightly, but not dramatically, as $K$ increases. We expect the same pattern to hold for the ensemble method, since it relies on the same subsampling method.
>
> **Q1:** Thank you for catching this typo. The self-regularization term is included to encourage minimizing the divergence between candidate posteriors, and so Equation (5) should have a plus sign. We have fixed this in the revised paper.
>
> **Q2:** While increasing diversity between ensemble members may be a good strategy when lots of training examples are available, it does not work well in the data-constrained setting. As we show in Lemma 2 in our paper, in the absence of any regularization the optimal (maximum likelihood) ensemble members are those that approximate a delta function at each training example (i.e. overfit to their respective subsamples). In this case, even two ensemble members trained on subsamples that differ by a single data point would have KL divergence approaching $\infty$, since one would assign near-zero probability mass to the example that differs. In short, encouraging diversity in the sparse-data regime encourages overfitting of each ensemble member.
>
> Fig. 12 illustrates this counterintuitive result empirically by varying both the regularization strength $\beta$ and the size of the failure dataset. We find that when the failure dataset is very small, a larger regularization strength ($\beta=10$) yields better performance, but smaller regularization strengths perform better when the failure dataset is large. We have added some discussion of this point in the appendix of our revised paper (page 20, immediately prior to Fig. 12).
>
> **Q3:** You are correct on the typo in Equation 3; we have fixed this in our revised paper. For $L_{cal}$, we have revised this equation to be more consistent in referring to $c$ as the variable and $c^*$ as its optimized value.
>
> *(continued in following comment due to space limitations)*

---

> > ### Author Response · Authors · 2024-11-22
> >
> > **Q4:** The ensemble model misses two key features that CalNF is able to infer. The first is the gap between the two blocks, with the ensemble model incorrectly having darker shading for cells (4, 5) and (5, 5), for example. The second is the fact that the block on the right is shifted downwards; the ensemble model shades the cells on the right too darkly in the 3rd row and too lightly in the 6th row. CalNF does better in inferring these two structural features of the ground truth.
> >
> > **Q5:** By data sensitivity, we are referring to the issue of overfitting (where the learned posterior is very sensitive to the exact data points included in the training set). Our method is designed to resist overfitting, and our experimental results indicate that it achieves improved performance on data-constrained learning problems (both in terms of the posterior learning results in Table 1 and the downstream anomaly classification results in Table 2).
> >
> > **Q6:** In lines 129-131, we claim that regularizing the KL divergence between the nominal and target distributions may not be helpful (even after tuning the regularization strength). Our reasoning is that many failures involve a distribution shift, so the distribution of parameters in failure examples is different from their distribution in nominal examples. When nominal and failure examples are drawn from different distributions, it does not make sense to artificially constrain the learned failure distribution to be close to the learned nominal distribution. This motivates the design of our method, where we do not explicitly regularize the learned failure distribution to be close to the learned nominal distribution. Instead, we apply regularization only between candidate failure distributions.
> >
> > Does that help answer your questions? We are happy to follow up if you have further questions or if there are any points that you are not satisfied with.

---

### Official Review · Reviewer_QvkQ · 2024-11-02

**Soundness:** 3
**Presentation:** 3
**Contribution:** 3
**Rating:** 6
**Confidence:** 3

**Summary:**

This paper presents Calibrated Normalizing Flows (CALNF), a method for modeling rare failures in data-limited settings by learning robust posterior distributions through self-regularization. CALNF combines nominal and limited failure data to capture event distributions effectively, demonstrated on benchmarks like UAV control and air traffic disruptions, including an analysis of the 2022 Southwest Airlines crisis.

**Strengths:**

1. The CALNF framework addresses an important challenge in failure modeling by using normalizing flows for posterior learning from limited data. This approach is innovative in combining data regularization with low-dimensional embedding optimization, which could be valuable in real-world failure analysis in autonomous systems.

2. The paper tests its methodology on practical datasets, including autonomous UAV control and air traffic disruptions. This inclusion strengthens the paper’s relevance for the conference theme, as it presents an approach applicable to practical, high-impact fields.

3. The paper compares CALNF against well-established baselines, such as KL and Wasserstein regularized methods, and ensemble methods. These comparisons and detailed analyses provide a fair view of CALNF’s effectiveness, especially in data-limited scenarios

**Weaknesses:**

1. The dual steps of the method—learning a low-dimensional embedding and calibrating a label space—add significant complexity, which may impact reproducibility and practical use, particularly in robotics where model simplicity and speed are often crucial. Additional clarifications or simplifications could improve accessibility to a broader audience

2. Although the authors acknowledge this in the Limitations section, the increased training time due to subsampling may limit CALNF's applicability to settings requiring rapid inference, such as real-time planning and decision-making in robotics. The authors could consider discussing strategies to mitigate this cost

3. While CALNF aims to model rare failures effectively, it does not provide direct risk estimates or probability bounds for these events. Adding a framework for estimating risk or failure probabilities could enhance the utility of this approach for preemptive failure detection and mitigation

4. The paper provides some analysis around sensitivity and regularization but lacks a comprehensive discussion of how well the approach generalizes to new data. For applications in robotics and autonomous systems, where new conditions are frequently encountered, more robust guarantees on generalization could significantly strengthen the work

**Questions:**

1. Could you provide more theoretical insights or empirical evidence on how CALNF handles data sparsity when transferring to new or unseen data points? For applications in autonomy and robotics, understanding the model's generalization capability could be essential.

2. How sensitive is CALNF to the choice of regularization hyperparameter β and the number of subsamples K? While some sensitivity analyses are included, it would be beneficial to understand how practitioners might adapt these parameters to different problem settings with minimal fine-tuning.

3. Given the added training cost of CALNF due to subsampling, how practical is this method for real-time applications in autonomy or robotics where immediate inference might be required? Would you recommend any adjustments for faster, low-latency performance?

---

> ### Author Response · Authors · 2024-11-22
>
> Thank you for your review and feedback! We have included responses to your questions below along with a revised version of our paper (with changes highlighted in blue).
>
> **Complexity & added training cost (Weaknesses 1 and 2, Question 3):** You ask about suitability to robotics applications. We would like to emphasize that while our method is more complex to train, there is no penalty at inference time compared to any other normalizing flow-based method. Because of this, we believe that our method would be suitable for real-time applications like robotics, where the normalizing flow trained using CalNF could be used for downstream tasks like anomaly detection or control. We have added some discussion of this point to the revised paper.
>
> We also acknowledge that our method is slightly more complex to implement than a vanilla normalizing flow (we provide an example implementation in the supplementary materials). As such, CalNF would not be the best choice for a problem with plentiful failure data, but our experiments (both comparisons and ablations) show that this additional complexity is needed to achieve good performance on data-constrained problems.
>
> **Sensitivity to hyperparameters $\beta$ and $K$ (Question 2):** Our results in Fig. 11 suggest that our method is not very sensitive to the choice of $K$, likely because there are so few data points (e.g. the SWI problem only includes 4 failure examples, so increasing $K$ just leads to sampling additional subsets that are statistically similar to those already sampled). Our results in Fig. 12 suggest that we may have been able to achieve better performance by tuning $\beta$ (e.g. higher $\beta$ seems perform better on very small failure datasets), but we wanted to avoid accidentally overfitting to our limited datasets through excessive hyperparameter tuning (the UAV and ATC datasets in particular only have a small amount of available real-world data).
>
> Practically, we fixed $K=5$ and $\beta=1$ for all of our experiments and found that we achieved good performance without further fine-tuning. We would recommend that practitioners start with these values, then potentially tune $\beta$ depending on how much failure data is available (with larger $\beta$ helping to avoid overfitting when very few failure examples are available).
>
> **Generalization to new data (Weakness 4 and Question 1):** You are correct that the ability to generalize beyond the training dataset is important. In fact, the inability of existing methods to generalize well based on limited failure datasets (due to overfitting) is the primary motivation for our method. In our experiments, we quantify how well our method generalizes to new data by reporting our model’s performance on a test set of previously-unseen failure examples. These empirical results (Tables 1, 2, and 3) show that our method generalizes well to previously-unseen failures.
>
> There is still an open question on how well our method would adapt to new failure examples from an entirely different distribution than what is seen during training, but this limitation applies to any learned generative model.
>
> **Risk estimates or probability bounds (Weakness 3):** We included this issue in the limitations section of our original paper. We agree with you that obtaining these risk estimates would be useful in many practical scenarios, and look forward to addressing the theoretical and practical challenges of obtaining accurate probability bounds from a very small number of examples in future work. If you have any thoughts or suggestions on this future direction, we would be happy to hear them!
>
> Does that help answer your questions? We are happy to follow up if you have further questions or if there are any points that you are not satisfied with.

---

> > ### Comment · Reviewer_QvkQ · 2024-11-26
> >
> > Thank you for responding to the questions. Most concerns are resolved properly. I am willing to consider to raise my score.

---

### Official Review · Reviewer_Wb4h · 2024-11-03

**Soundness:** 3
**Presentation:** 3
**Contribution:** 3
**Rating:** 6
**Confidence:** 3

**Summary:**

This paper aims to address rare event modeling problems where failure data is in a very limited amount. Specifically, the authors propose a generative modeling framework called Calibrated Normalizing Flows (CALNF) to learn a shared representation from both normal and rare failure data. Furthermore, they leverage calibration and self-regularization processes to avoid overfitting. They provide empirical results on benchmarks and real-world cases.

**Strengths:**

1. The idea is intuitive and easy to understand.
2. The paper is well-organized and easy to follow.
3. The qualitative visualizations on benchmark datasets are impressive.
4. The case study on the 2022 Southwest Airlines scheduling crisis is persuasive.

**Weaknesses:**

1. The authors assume that failure events are not entirely disjoint from normal events, however, this might not hold in cases where failures are out-of-distribution (OoD) data. This means the failures can have drastically different characteristics from the normal ones. Can the proposed method still represent these rare events accurately in this case?
2. Even with self-regularization, the model’s performance might still be sensitive to the selection of the number of subsamples $K$ and regularization weights $\beta$. What method did you use to select $K$ and $\beta$? Was it tuned manually, or is there a heuristic that works well across different failure datasets?
3. As discussed by the authors, the proposed method introduces computation overhead. Have you considered how to extend the proposed method to high-dimensional data such as images?

**Questions:**

See Weaknesses for the major concerns. Other questions:
1. Is the calibrated normalizing flows model the optimal solution? Can the proposed model generalize to other generative models such as GAN and Diffusion models?

---

> ### Author Response · Authors · 2024-11-22
>
> Thank you for your review and feedback! We have included responses to your questions below along with a revised version of our paper (with changes highlighted in blue).
>
> 1. **Cases when failure events are drastically different from nominal events:** This is an interesting question! The implicit assumption behind our work is that there is a benefit from learning a shared representation for both the nominal and failure events. In Theorem 1, we show that our method provides an implicit regularization between the nominal and failure events. This could be either a benefit or a drawback (depending on whether there is actually some structural similarity between the nominal and failure distributions). We have added a remark in the limitations section of our paper discussing this point.
> 2. **How did we select $K$ and $\beta$?** We did not spend much time tuning either parameter: we used $K=5$ and $\beta=1$ on all problems. Our results in Fig. 11 suggest that our method is not very sensitive to the choice of $K$, likely because there are so few data points (e.g. the SWI problem only includes 4 failure examples, so increasing $K$ just leads to sampling additional subsets that are statistically similar to those already sampled). Our results in Fig. 12 suggest that we may have been able to achieve better performance by tuning $\beta$ (e.g. higher $\beta$ seems perform better on very small failure datasets), but we wanted to avoid accidentally overfitting to our limited datasets through excessive hyperparameter tuning (the UAV and ATC datasets in particular only have a small amount of available real-world data). As a result, we kept $K$ and $\beta$ fixed in our experiments and found that to be sufficient.
> 3. **Extending the proposed method to high-dimensional data?** The main bottleneck in expanding to high-dimensional data (e.g. images) comes from evaluating the underlying normalizing flow. To address this issue, we can leverage existing work on adapting normalizing flows for image data, such as the Glow architecture (Kingma & Dhariwal 2018). In Section 4.5 of our paper, we demonstrate applying our method to both the MNIST and CIFAR-10 image datasets using the Glow architecture for the underlying flow. More work is certainly needed to improve on these results on image modeling, but we hope that these experiments demonstrate how our method might be adapted to high-dimensional image data.
> 4. **Are normalizing flows the optimal solution?** This is a good question. The general idea of our paper (self-regularization of an ensemble model) could likely be applied to other generative models, but our use of KL divergence to regularize between candidate posteriors means that we need to be able to evaluate the learned likelihood $q_\phi$ for different labels $c$ in addition to drawing samples from the learned distribution. Since evaluating the likelihood for GANs and diffusion models is not straightforward, normalizing flows are a natural choice for our method. There are also other benefits to normalizing flows for downstream tasks; for example, the ability to evaluate the likelihood allows them to be adapted easily to anomaly classification applications.
>
> Does that help answer your questions? We are happy to follow up if you have further questions or if there are any points that you are not satisfied with.

---

### Official Review · Reviewer_MKpx · 2024-11-04

**Soundness:** 2
**Presentation:** 2
**Contribution:** 3
**Rating:** 8
**Confidence:** 2

**Summary:**

In this work, the authors propose an algorithm for rare-event modeling and failure detection using normalizing flows. The motivation stems from the observation that rare events are significantly low in number, making traditional approaches like prior regularization or simple bootstrapping ineffective. To address this challenge, the authors develop an optimization pipeline that trains a normalizing flow with a built-in calibration mechanism, which identifies the optimal latent factors associated with a target (failure) dataset. The paper is supported by extensive theoretical analyses, experiments, and an interesting case study, all demonstrating the efficacy of the proposed approach.

**Strengths:**

The paper addresses an important and intriguing problem in modeling rare events or failures with generative models, which often struggle to generalize when trained on limited data. It is mathematically well-supported and includes relevant case studies and experiments on standard benchmarks

**Weaknesses:**

Refer to Questions

**Questions:**

* How would the performance change if the calibration procedure is conducted jointly with the training of the normalizing flow? Would this make the optimization process more challenging?
* Figure 4 needs to be referenced in the text of the paper for clarity.
* Can this framework be extended to problems where the calibration label is continuous-valued? Understanding this extension would be useful, as many scientific applications involve real-valued outputs or measurements.
* Could the authors elaborate on the anomaly rejection and few-shot inference experiment details? A more comprehensive explanation of these procedures would enhance the clarity and depth of the paper.

---

> ### Author Response · Authors · 2024-11-22
>
> Thank you for your review and feedback! We have included responses to your questions below along with a revised version of our paper (with changes highlighted in blue).
>
> 1. **Calibrating at the same time as training:** Since the loss function used for calibration ($L_{cal}$) is the same as the third term in the loss function used for training the normalizing flow ($L$), it would be equivalent to optimize $c^*$ and the parameters of the flow jointly. In Algorithm 1, we put the model and calibration updates on two separate lines for clarity, but they could be implemented as one joint update. We have added some discussion to this effect.
> 2. **Referencing Fig. 4:** thank you for catching this mistake. We now discuss Fig. 4 on page 6 of the revised paper.
> 3. **Real-valued calibration label:** Our method fully supports continuous values for the calibration label $c^*$, the observations $x$, the latent variables $z$, and the context variables $y$, and all of our experiments use continuous values for these variables. We have added a note on this to the problem statement in our paper.
> 4. **Additional details on anomaly rejection and few-shot image learning experiments:** Thank you for this feedback. We have added additional information on these experiments to our revised paper.
>
> Does that help answer your questions? We are happy to follow up if you have further questions or if there are any points that you are not satisfied with.

---

> > ### Comment · Reviewer_MKpx · 2024-11-26
> > **Post Rebuttal**
> >
> > I thank the authors for providing responses to each of my questions.  I think they have been clarified. I will be increasing my score.

---

### Author Response · Authors · 2024-11-22

We would like to thank all of our reviewers for their careful reviews and helpful feedback. We have responded to each reviewer individually, and we look forward to a constructive discussion.

---

### Author Response · Authors · 2024-12-01
**Summary of review period**

As the discussion period comes to a close, we would like to thank our reviewers once again for their constructive feedback. Regardless of the decision, we believe that your feedback has lead to a higher-quality revised paper. We have responded to each reviewer individually, and we would like to summarize a few major points here.

**Clarification on suitability for real-time use:** Several reviewers asked how our method would adapt to use in real-time applications. We clarified, both in our responses below and in our revised paper, that our method has *no inference-time penalty* relative to other normalizing flow-based methods (the only additional cost to our method is at training time). While our primary focus has been on the offline "failure analysis" case, motivated by the novel Southwest airlines case study in our paper, we believe that our method could easily be adapted to real-time use for downstream tasks like anomaly detection.

**Clarification on hyperparameters:** Several reviewers asked about the sensitivity of our method to the hyperparameters $K$ and $\beta$. We have included sensitivity experiments in the appendix of our paper, and we provided the additional clarification that we held these hyperparameters constant across experiments rather than optimizing them for each problem. Our sensitivity results (Fig. 12) suggest that we could have achieved even better performance on some problems by tuning these parameters, we chose not to do so to avoid the risk of overfitting to the small amount of validation data available on some problems.

**Clarification of performance advantage relative to baselines:** One reviewer points out that the performance gap between our method and one baseline is relatively small on some problems and asks if the additional complexity of our method is worth it. In response, we provided additional experimental data during the discussion phase (an ablation of the relevant baseline to demonstrate that, like our method, it also needs access to nominal data to achieve good performance), and we explained that while the performance gap is small on some problems when measured by the ELBO metric (Table 1, where even small differences in ELBO lead to large changes in qualitative performance in Fig. 3), the performance gap is much larger and more consistent when measured by performance on downstream tasks like anomaly detection (Table 2) and image generation (Table 3 and Fig. 6). While one reviewer (mQHM) comments that performance on these downstream tasks is not relevant to the main claims of our paper, we politely disagree --- posterior learning is a foundational capability for a range of downstream tasks, and practitioners will prioritize methods that perform well on these downstream tasks. As a result, we believe that these results demonstrate that our method presents a more reliable approach to posterior learning on highly data-constrained problems, not only outperforming baselines on many problems but also providing a better foundation for downstream tasks like failure analysis, anomaly detection, and failure example generation.

We hope that we have addressed all of our reviewers' concerns. If there are any points where a reviewer would like additional clarification or discussion, we would be happy to engage during the remainder of the review process.

---

### Meta-Review · Area_Chair_VtiJ · 2024-12-19

**Metareview:**

The paper introduces a method called Calibrated Normalizing Flows (CalNF) to model rare events under severe data constraints. With a novel self-regularization strategy, they enable failure modeling for limited data settings and inverse problems, with a real world application in root cause analysis for the 2022 Southwest Airlines crisis. The reviewers agree that the problem is relevant and the method is well supported theoretically. The main concerns of the reviewers were: additional complexity at test time for real-time use and the performance advantage over baselines. Overall, the reviewers agree that this is a solid paper.

**Additional Comments On Reviewer Discussion:**

During the discussion, two reviewers raised their score, and one reviewer raised their confidence with an already positive score.

---

### Decision · Program_Chairs · 2025-01-22

Accept (Poster)